# Quorum Quenchers from *Reynoutria japonica* in the Battle against Methicillin-Resistant *Staphylococcus aureus* (MRSA)

**DOI:** 10.3390/molecules28062635

**Published:** 2023-03-14

**Authors:** Maliha Fatima, Arshia Amin, Metab Alharbi, Sundas Ishtiaq, Wasim Sajjad, Faisal Ahmad, Sajjad Ahmad, Faisal Hanif, Muhammad Faheem, Atif Ali Khan Khalil

**Affiliations:** 1Department of Biosciences, Capital University of Science and Technology, Islamabad 44000, Pakistan; 2Department of Pharmacology and Toxicology, College of Pharmacy, King Saud University, P.O. Box 2455, Riyadh 11451, Saudi Arabia; 3Department of Biological Sciences, National University of Medical Sciences, Rawalpindi 46000, Pakistan; 4National Center for Bioinformatics, Quaid-i-Azam University, Islamabad 45320, Pakistan; 5Department of Health and Biological Sciences, Abasyn University, Peshawar 25000, Pakistan; 6Department of Computer Sciences, Virginia Tech, Blacksburg, VA 24060, USA; 7Department of Microbiology Military Hospital, National University of Medical Sciences, Rawalpindi 46000, Pakistan; 8Department of Pharmacognosy, Institute of Pharmacy, Lahore College for Women University, Lahore 54000, Pakistan

**Keywords:** quorum sensing, quorum quenching, resveratrol, pharmacokinetics, multidrug resistance

## Abstract

Over the past decade, methicillin-resistant *Staphylococcus aureus* (MRSA) has become a major source of biofilm formation and a major contributor to antimicrobial resistance. The genes that govern biofilm formation are regulated by a signaling mechanism called the quorum-sensing system. There is a need for new molecules to treat the infections caused by dangerous pathogens like MRSA. The current study focused on an alternative approach using juglone derivatives from *Reynoutria japonica* as quorum quenchers. Ten bioactive compounds from this plant, i.e., 2-methoxy-6-acetyl-7-methyljuglone, emodin, emodin 8-o-b glucoside, polydatin, resveratrol, physcion, citreorosein, quercetin, hyperoside, and coumarin were taken as ligands and docked with accessory gene regulator proteins A, B, and C and the signal transduction protein TRAP. The best ligand was selected based on docking score, ADMET properties, and the Lipinski rule. Considering all these parameters, resveratrol displayed all required drug-like properties with a docking score of −8.9 against accessory gene regulator protein C. To further assess the effectiveness of resveratrol, it was compared with the commercially available antibiotic drug penicillin. A comparison of all drug-like characteristics showed that resveratrol was superior to penicillin in many aspects. Penicillin showed a binding affinity of −6.7 while resveratrol had a score of −8.9 during docking. This was followed by molecular dynamic simulations wherein inhibitors in complexes with target proteins showed stability inside the active site during the 100 ns simulations. Structural changes due to ligand movement inside the cavity were measured in the protein targets, but they remained static due to hydrogen bonds. The results showed acceptable pharmacokinetic properties for resveratrol as compared to penicillin. Thus, we concluded that resveratrol has protective effects against *Staphylococcus aureus* infections and that it suppresses the quorum-sensing ability of this bacterium by targeting its infectious proteins.

## 1. Introduction

Antibiotic resistance has created an alarming situation in the health sector globally and antibiotic-resistant pathogens are often described as “superbugs” [1]. Repeated and uncontrolled use of antibiotics to target DNA, RNA, and protein synthesis exerts bacteriostatic or bactericidal effects on multiple targets, resulting in strong selective pressure on bacterial communities which subsequently gives rise to bacterial strains resistant to those antibiotics. Virulence, pathogenicity, and biofilm formation by resistant pathogens are significant problems that create unusual medical emergency situations. The main representative among these notorious pathogens is methicillin-resistant *Staphylococcus aureus* (MRSA) [2]. *S. aureus* has quickly become a leading cause of healthcare-related diseases. The COVID-19 pandemic was associated with a 13% increase in MRSA infections in 2020 in the Americas compared to 2019. MRSA has been linked to two types of infection: hospital-acquired and community-acquired (CAI) [3]. Persistent infections associated with biofilm development can persist in host tissues and implanted materials such as bone, catheters, pacemakers, and prosthetic joints, resulting in osteomyelitis, heart valve endocarditis, and other complications [4]. Many factors have been implicated in biofilm development, such as bacterial density, stress responses, physiological features, antibiotic resistance, neutralization of antibiotics by EPSs (exopolysaccharides), enzyme synthesis, and QS (quorum-sensing) capabilities [5].

The failure of several conventional pharmaceutical approaches and continuous increase in mortality statistics has created an urgent need for new approaches. Alternative strategies to the use of antibiotics for bacterial infections are based upon the quenching of signaling mechanisms. Disruption of these pathways may play an important role in controlling microbial gene expression in human infections [3]. These signaling pathways are responsible for inter- and intraspecies communication and for the regulation of gene expression, and are categorized broadly under the convenient term “quorum-sensing” [4]. Bacterial cell-to-cell connection has garnered attention in recent years as studies have demonstrated the function of quorum signals in the adhesion and proliferation of harmful bacteria [5]. The discovery of quorum signals has introduced a new dimension to the compounding health crises. Bacterial communication is carried out via the synthesis of tiny signal molecules known as autoinducers. The most thoroughly researched autoinducer molecule in Gram-positive bacteria is the autoinducer peptide (AIP) [6]. The accessory gene regulator (Agr) locus is a significant regulator in the *S. aureus* QS system, consisting of two operons controlled by the P2 and P3 promoters. The P2 operon controls AgrA, -B, -C, and -D synthesis in response to extracellular AIP [7]. The P3 operon controls RNAIII expression, which has been shown to be responsible for the transition from a sticky phenotype to a poisonous one [8]. Quorum sensing regulates important bacterial behaviors, e.g., attachment to surfaces, biofilm formation, bioluminescence, the secretion of different types of chemicals, motility, virulence, and pathogenicity [9]. Several synthetic and natural substances have been explored for their quorum-sensing inhibitory action [10]. However, the limitations of these drugs against various forms of resistance have prompted a quest for new quorum-sensing inhibitors for possible use in a variety of applications.

Considering the rise of antibiotic resistance, QS inhibitors may provide an alternative to standard antibiotic therapies. Bioactive phytoconstituents are being utilized to treat infectious diseases caused by biofilmogenic bacteria. Plant-based bioactive compounds can decrease the expression of disease pathogenesis genes by interacting with QS-associated virulence factors and affecting biofilm formation. Several compounds have previously been shown to have antibiofilm activities, including quercetin, catechin, rosmarinic acid, limonoid, ichangin, apigenin, kaempferol, and naringenin [11]. It is critical to identify powerful QS inhibitors (QSIs), ideally from natural sources. Plant secondary metabolites may result in the effective treatment of a variety of illnesses [11].

In this study, resveratrol, a typical stilbenoid commonly utilized in dietary supplements and renowned for its antioxidant potential, was evaluated for a QS inhibitory effect [12]. Resveratrol has been evaluated for its ability to inhibit various QS-regulated behaviors of infectious pathogens, namely biofilm formation, exopolysaccharide synthesis, and motility [13]. We investigated the antibacterial compound isolated from *Reynoutria japonica*. *S. aureus* was used as a reference organism and its virulence protein targets were docked against multiple ligands. Hit molecules were selected based on their physiochemical and pharmacokinetic properties. To explore the structural changes in the AgrC receptor protein of *S. aureus*, in silico research comprising molecular docking and simulation studies was performed to better understand the mechanism of the QS inhibitory function. Our study reveals that the antibacterial properties of resveratrol are better than those of penicillin in many aspects, and that resveratrol has the potential to suppress the quorum-sensing activity of bacteria [14]. 

## 2. Results

The FASTA sequences of the accessory gene regulator proteins A, B, and C and the signal transduction protein TRAP were retrieved from UniProt under accession numbers P0A017, P0C1P7, O07911, and Q84DC6 and were 238, 189, 430, and 167 residues in length, respectively (Figure 1). To inhibit the biosynthetic pathways of these protein targets, a molecular docking approach was used followed by molecular dynamic simulations. The Agr system is a global staphylococcus regulator with a dual regulatory effect on staphylococcal virulence. In aspects of clonal lineage distribution, antibiotic resistance profile, biofilm generation, and virulence factor expression, Agr groupings differ [15]. Because of its role in modulating virulence factor production and biofilm development, the agr system is an interesting therapeutic target. Interfering with or totally suppressing the agr system might be a useful method for lowering staphylococcal pathogen virulence and controlling staphylococcal disease. Additionally, because AgrC catalyzes AgrA phosphorylation and activation, inhibitors inhibiting AgrC or AgrA may be effective at preventing disease development [16]. AgrB is the most unique component of the staphylococcal Agr system, since its sequence varies in comparison to other quorum-sensing proteins. The N-terminal domain of AgrB is usually inherited in staphylococcal species, whereas the initial 34 residues of the first transmembrane hydrophilic domain are fully conserved across the four *S. aureus* Agr types [17]. 

### 2.1. Physiochemical Characterization of Proteins

ProtParam was used for the prediction of different parameters, including both physical and chemical properties, of the selected protein targets. These characteristics were used to compute and assess the molecular weight, composition of amino acids, theoretical protein index value, atomic protein composition, extinction coefficient, estimated half-life of protein instability, aliphatic index, and grand average of hydropathicity. A PI of more than 7 implies that a protein is basic, while a PI of less than 7 indicates that it is acidic. Light absorption is represented by the extinction coefficient. An index value below 40 indicates protein stability, while an index value greater than 40 indicates protein instability (Table 1).

### 2.2. 3D Structural Prediction of Proteins

The 3D structures of accessory gene regulator proteins A, B, and C and the signal transduction protein TRAP were taken from Alphafold under UniProt IDs P0A017 (crystal structure available with 1.6 Å resolution) and P61637, Q2FWM5, and Q2FFR1 (less than 2.0 Å resolution as confirmed via Ramachandran plot) [18]. The protein structures were prepared in PyMOL by removing water molecules and ligands [19]. After the removal of ligands and other atoms, the missing polar hydrogens were added. Energy minimization for the structures was performed to achieve stable conformation by preventing overlaps, as shown in Figure 2. Dihydrofolate reductase (DHFR) is an enzyme associated with AgrA and AgrB which is responsible for the regulation of reduced folate pools, which are required to produce purines, thymidylate, methionine, glycine, pantothenic acid, and N-formyl-methionyl tRNA. Inhibition of DHFR causes tetrahydrofolate depletion, and, eventually, cell death. DHFR has received extensive attention as an antibacterial agent target [20]. Patients, physicians, and public health organizations are all concerned about the spread of methicillin-resistant *Staphylococcus aureus* (MRSA) in hospitals and communities. The *S. aureus* DHFR shows preservation of the conserved fold seen in previously released crystal structures of DHFRs from other species, with eight strands, a sheet, and four helices comprising the substrate and cofactor binding sites [17].

### 2.3. Functional Domain Identification of Proteins

The Interpro database (https://www.ebi.ac.uk/interpro/ accessed on 15 October 2022) was used to determine the domains and functional locations of the proteins. Accessory gene regulator protein A is a 238aa long protein consisting of two domains. One is the Lyt-TR DNA-binding domain, starting from residue 143 and ending at 238, while the other is the receiver domain, starting at residue 1 and ending at 125. The AgrB protein is a 186aa long protein consisting of a single domain called the accessory gene regulator B domain, starting at residue 6 and ending at 186. The AgrC protein, with a sequence of 430aa, consists of the sensor histidine kinase NatK [21], a C-terminal domain starting at residue 325 and ending at 427. The signal transduction protein TRAP, with a sequence of 167aa, consists of a single domain called the antibiotic biosynthesis monooxygenase domain, which starts at residue 67 and ends at 158 (Figure 3). 

AgrC and AgrA form a two-component signal transduction system, with AgrC acting as a membrane histidine kinase and AgrA acting as a response regulator. The cytoplasmic membrane contains AgrB, a 22 kDa peptidase responsible for AgrD proteolysis. It comprises six transmembrane segments, four of which are hydrophobic helices and two of which are hydrophilic loops containing many positively charged amino acid residues. DHFR has been employed in various therapeutic settings as resistance to antimicrobial drugs has become common [17].

### 2.4. Ligand Selection

Ligands were retrieved from the chemical information database PubChem (https://pubchem.ncbi.nlm.nih.gov accessed on 10 October 2022). After the selection of ligands, energy minimization was carried out using Chem Pro software (Chem3D v. 12.0.2) [22]. All ligands except hyperoside and coumarin obeyed the Lipinski rule of five. The selected ligands, along with molecular formulas, molecular weights, and chemical structures, are represented in Table 2.

### 2.5. Molecular Docking

The docking study was performed using accessory gene regulator proteins A, B, and C and TRAP and the ligands 2-methoxy-6-acetyl-7-methyljuglone, emodin, emodin 8-o-b glucoside, polydatin, resveratrol, physcion, citreorosein, quercetin, hyperoside, and coumarin. The ligands with the best binding score values with the target proteins are presented in Table 3. The current study adopted the protocol for AutoDock Vina v4.2, including the ligand and protein pdbqt files with the docking grid set at 30 Å × 30 Å × 30 Å [23]. The grid was centered at x, y, and z dimensions of 12.020, 4.545, and 36.451, respectively. Selected ligand molecules were docked to the active sites of the targets using AutoDock Vina. The highest score of −9.9 kcal/mol was achieved for the compound emodin 8-o-b glucoside, and the respective binding affinities for the top 10 compounds are provided in Table 3. Detailed visualization analysis was carried out through UCSF Chimera v1.16 and used to determine the preferred ligand binding orientations.

### 2.6. Active Site Identification

To identify the active sites of the proteins, Computed Atlas of Surface Topography of proteins (CASTp) software v3.0 (http://sts.bioe.uic.edu/castp/index.html?2r7g accessed on 10 October 2022) was used. This software predicts available pockets for binding and provides insights about the surface area and volume of pockets. The active sites of accessory gene regulator proteins A, B, and C and TRAP are shown in red in Figure 4.

### 2.7. Interaction of Ligands and Target Proteins

The interactions of the ligands and the active pockets of the proteins were calculated in order to interpret the docking results. Hydrogen bonding and hydrophobic bonding interactions were studied using Ligplot plus (version v.1.4.5) [24] (Table 4). The results showed the binding forces among the residues and atoms of the ligands, along with multiple hydrogen bonds and their distances. Most hydrogen bonds interacted with the serine residue cloud of the target molecules, as shown in Table 4 and Table 5.

### 2.8. Ligands’ ADMET Properties

Lipinski’s rule of five was employed as a preliminary step to determine actual bioavailability and artificial availability. A second investigation was carried out involving calculation of the ADMET characteristics of ligands as a measure of pharmacokinetic properties using the online application pkCSM (https://biosig.lab.uq.edu.au/pkcsm/ accessed on 8 October 2022). Water solubility and skin absorption for all ligands were low, while CaCO_2_ permeability was normal. Intestinal absorption rates of juglone, physcion, and coumarin were more than 90%, while this rate was average for emodin and resveratrol and low for the remaining ligands. Skin permeability for all ligands was low. Juglone showed a negative p-glucoprotein substrate value, while all other ligands showed a positive value for a single factor. If a compound binds to a Pgp substrate, it may be quickly pumped out of cells, lowering its absorption (Table 6).

### 2.9. Distribution, Metabolic, and Excretion Properties of Ligands

The transport of drugs from one region to another within the body was investigated. In humans, the dispersion (VDss, defined as log L/kg) is one of the four ADMET properties; the others are the fraction unbound in humans (Fu), the permeability of the blood–brain barrier (BBB) expressed as log BB, and the permeability of the central nervous system expressed as log PS. The VDSS values of all ligands were low, while the Fu values of all ligands were positive. The BBB permeability values of all ligands were in the range of −1. The log PS values of emodin 8-o-b glucoside, polydatin, citreorosein, quercetin, and hyperoside were less than -3, while for the other ligands this value was greater than −3 (Appendix A). Cytochrome P450, also known as CYP1A2, CYP2C19, CYP2C9, CYP2D6, and CYP3A4, is an essential cleaning enzyme present in the liver. The metabolic properties of the ligands are presented in Appendix A. The kidneys are involved in drug excretion through their important functions in glomerular filtration and biliary excretion. Narcotics can also be eliminated through perspiration, saliva, and tears. Total clearance represented as log (CL tot) in ml/min/kg is one model of excretion property, and renal OCT2 substrate can predict outcomes as Yes/No (Appendix A).

### 2.10. Ligand Toxicity 

The maximum tolerated dose (MRTD) determines the toxicity of a hazardous substance in an individual. This information aids in directing a treatment regimen’s initial indicated dosage in phase 1 clinical trials. The MRTD is represented logarithmically (log mg/kg/day). A chemical has a low MRTD if its value is less than or equal to 0.477 log (mg/kg/day) and a high MRTD if its value is greater than 0.477 log (mg/kg/day). The maximum tolerated doses of juglone, resveratrol, quercetin, and hyperoside were high. All ligands showed no hERGI or hERGII inhibition. Hepatotoxicity was shown only by 2-methoxy-6-acetyl-7methyljuglone, and no ligand showed skin sensitivity. No ligand showed *T. pyriformis* activity less than −0.5 log μg/L. The minnow toxicity values of all ligands were greater than 0.5 mM, which is considered safe (Appendix A).

### 2.11. Lipinski Rule of Five

The Lipinski rule was applied to our analysis of different ligands from *Reynoutria japonica*, as shown in Table 7.

Table 7 shows the molecular weights, log*p* values, and hydrogen bond acceptor and donor values of the ligands from *Reynoutria japonica*. A compound is considered an acceptable drug if it follows three or more rules, and is considered poorly absorbed if it violates two or more rules. With the exception of hyperoside and coumarin, nearly all the ligands followed the Lipinski rule of five.

### 2.12. Lead Compound Identification

Physiochemical and pharmacokinetic properties determine the final destiny of a compound as a drug or nondrug. Emodin 8-o-b glucoside, polydatin, hyperoside, and coumarin did not obey the Lipinski rule of five and so were removed in the primary screening. Based on the binding score, ADMET properties, physiochemical properties, and Lipinski rule of five, resveratrol was selected as the lead compound which could inhibit the target proteins.

### 2.13. Comparative Investigation of Lead Compound vs. Penicillin 

The comparison between penicillin and resveratrol helped us to identify a better treatment for infectious diseases. The comparison was performed using different parameters, including the ADMET properties and physiochemical properties of both compounds. Penicillin was selected as a reference drug because of its repeated use and effectiveness against bacterial infections. It is used to treat infections caused by Gram-positive bacteria, especially staphylococcal and streptococcal infections. Due to its low oral absorption, it is given intravenously or intramuscularly. Natural penicillin can be used as a first- or second-line antibiotic against Gram-positive bacteria. Patterns of resistance, susceptibility, and treatment options differ by region reference. ADMET properties include values regarding to drug absorption, distribution, metabolism, excretion, and toxicity. These values helped us to determine the drugs’ activity and efficiency.

### 2.14. Comparison of Absorption Properties 

The absorption properties of penicillin and resveratrol were compared (Table 8).

The water solubility, skin permeability, and intestinal absorption values of resveratrol were higher than those of penicillin.

### 2.15. Comparison of Distribution Properties 

The distribution properties of penicillin and resveratrol were compared (Table 9).

The distribution properties of the bioactive compound resveratrol were better than those of the drug penicillin.

### 2.16. Comparison of Metabolic Properties 

The metabolic properties of penicillin and resveratrol were compared (Table 10).

The CYP-3A4 substrate was found in both resveratrol and penicillin, but the CYP1A2 inhibitor was present only in resveratrol, which may help in the metabolism of the drug.

### 2.17. Comparison of Excretion Properties 

The excretion properties of penicillin and resveratrol were compared (Table 11).

The total clearance value of resveratrol in the body was greater than that of penicillin, indicating superior excretion of the drug from the body.

### 2.18. Comparison of Toxicity 

The toxicity parameters of penicillin and resveratrol were compared. The maximum tolerated dose was 1.284 for penicillin and 0.561 for resveratrol, and the oral acute toxicity rate of resveratrol was greater than that of penicillin (Table 12).

### 2.19. Lipinski Rule of Five

Penicillin and resveratrol were compared in terms of the Lipinski rule of five (Table 13).

It was found that resveratrol showed better results than penicillin in terms of log*p* value and hydrogen bond donors and acceptors.

### 2.20. Docking Score Comparison

The lead compound, resveratrol, showed a higher Vina score than the standard drug penicillin (Table 14).

The above results suggest that the ADMET properties and docking score of resveratrol are better than those of penicillin, so resveratrol can be used as an antibacterial compound in future therapeutic applications.

### 2.21. Molecular Dynamic Simulations 

To better characterize the enzyme–inhibitor complexes, MD simulations were used. These simulations emphasize residues’ binding affinities and display the dynamic behavior of proteins. The inhibitor molecules emodin 8-o-b glucoside, hyperoside, penicillin, and resveratrol in complexes were investigated using molecular dynamic simulation over a time frame of 100 ns. To unravel a molecule’s functional variability, a comprehensive understanding of its structure is needed. The simulation trajectories were first evaluated using the root mean square deviation (RMSD) based on all the carbon alpha atoms of the complexes. As can be seen in Figure 5, all the systems displayed stable dynamics except that of penicillin. No major deviations were reported, which indicates that the intermolecular interactions between the biomolecules and ligands were quite stable, as shown in Figure 6. 

The average RMSD value for the docked complexes, i.e., resveratrol, emodin 8-o-b glucoside, and hyperoside, was 1 Å, with maximum peaks of 1.67 Å, 1.3 Å, and 1.24 Å, respectively (Figure 5). The ligands were well positioned inside the binding regions with a slight to and fro motion (Figure 5). Meanwhile, the penicillin system gained some stability towards the end of the simulation but showed deviations greater than 3 Å.

To assess the structural compactness as a time function for the 100 ns simulations of the protein–ligand complexes, the radius of gyration was determined. Similar findings were revealed throughout the simulation time frame, showing a stable environment across the whole run with a mean square value of ≥20 Å. The analysis showed that the systems maintained a compact nature throughout the simulation period and no major conformational changes were noted. The radius of gyration plots for the systems can be seen in Figure 7.

Root mean square fluctuation (RMSF) analysis was carried out to obtain information on residue level flexibility and stability. The average RMSF values of emodin 8-o-b glucoside, penicillin, resveratrol, and hypersoide were 1.1 Å, 1.6 Å, 1.8 Å, and 1.2 Å, respectively (Figure 8). 

## 3. Discussion

In recent years, bacteria have become more resistant to various treatments, including by adopting new survival strategies and modifying their motility, virulence, and pathogenicity patterns. The main contributing factor to bacterial antibiotic resistance is the repeated misuse of antibiotics, which, along with other environmental factors, is creating multidrug-resistant pathogens and presenting serious challenges for infection management [2]. The failure of antibiotics has forced researchers to search for alternative approaches and novel ways to tackle resistance. As a result, a new research focus is the disruption of bacterial communication channels so that pathogenic and virulent traits cannot be transferred [25]. Bacterial communication is density-dependent and uses signals known as quorum sensing [26]. Scientists are currently attempting to find ways to disrupt this signaling mechanism. This new strategy of suppression of quorum sensing is called quorum quenching, which can be achieved at the level of signal production, signal reception, or signal transduction [27]. The current investigation aimed to use computational methods to discover a novel, nontoxic, and natural antibacterial compound for the treatment of infectious diseases that could be used in the near future as an efficient drug. The medicinal plant used in this study was Japanese knotweed, *Reynoutria japonica*. This plant was selected because it has been shown to have exceptional antipathogenic activities against several diseases [10]. It contains approximately 92 phytochemicals from the quinone, flavonoid, stilbene, coumarin, and lignin families that have been employed for centuries in over 100 Chinese medicinal treatments for a variety of ailments [28,29]. Keeping in mind this therapeutic potential, data-mining studies were performed and the 10 best ligands were selected. The target proteins of *Staphylococcus aureus* were selected based on their infectious properties: accessory gene regulator protein A, accessory gene regulator protein B, accessory gene regulator protein C, and the signal transduction protein TRAP [30]. These proteins were investigated for their physiochemical properties, domain identification, and binding pockets using different algorithms. The FASTA sequences of these proteins were retrieved from UniProt and the 3D structures were retrieved from Alphafold. Ligands were prepared and filtered for their drug-like properties. This was followed by application of a molecular docking protocol to check the binding affinities that led to the formation of hydrogen bonds and other linkages, including hydrophobic interactions. After detailed analysis of the ADMET properties and docking scores, the four best-scoring compounds, i.e., 2-methoxy-6-acetyl-7-methyljuglone, emodin, resveratrol and physcion, were identified as hit molecules. Resveratrol was identified as the lead compound based on its binding affinity to accessory gene regulator protein C. According to the literature, resveratrol is a naturally occurring polyphenolic antioxidant that belongs to the stilbene family [13]. It can suppress bacterial and fungal growth, modify the expression of virulence factors, diminish biofilm formation, and impact the sensitivity of bacteria to several classes of conventional antibiotics. It increases the effectiveness of aminoglycosides against a variety of Gram-positive bacteria [31]. Many health benefits, including antioxidant, anti-inflammatory, and immunomodulatory effects, and improvements in the symptoms of cancer, liver diseases, diabetes, obesity, Alzheimer’s disease, and Parkinson’s disease, are also associated with resveratrol [32]. More research is needed to explore its exact mechanisms of action, as well as its impact on the human body and any safety concerns. The current research was based on the novel approach of targeting the quorum-sensing system. The Agr locus, which regulates a wide range of virulence determinants in addition to metabolic genes, is primarily responsible for quorum-sensing regulation in *Staphylococcus aureus*. Agr has a major influence on several forms of staphylococcal illness. Agr typically promotes pathogenesis in acute diseases by boosting the production of aggressive virulence factors such as toxins and degradative exoenzymes [33]. In contrast, Agr plays a more complex function during chronic infections, as mutations in Agr result in greater biofilm formation but lower ability to disperse, and are associated with improved patient outcomes in persistent bacteremia [34,35]. In the case of the Agr quorum-sensing system, accessory gene regulators B and D are involved in the production of autoinducer peptides. Accessory gene regulator C receives signals when these peptides reach a threshold concentration, while accessory gene regulator A acts as transducer and upregulates virulence and biofilm-forming factors. If resveratrol can act as a quorum quencher by binding with AgrB and -D and altering their structures, then these defective proteins will not be able to produce signals and communication will be disturbed. If quenching is to be done at level of signal reception, then the target protein will be AgrC and changes in its structure will make it unreceptive to signals. The third alternative involves changing AgrA, which will decrease the control of enterotoxins, alpha-toxins, leucocidins, degradative exoenzymes, and phenol-soluble modulins which interact with virulence and pathogenicity-producing factors [36]. Variations in the sequences of AgrB, AgrC, and AgrD result in the creation of AIPs with varying signaling specificities, allowing for self-activation and cross-inhibition of nonself Agr groups [37]. The inhibitor molecules were evaluated after docking using molecular dynamic simulations. The simulations highlighted the binding affinities of the residues and demonstrated the dynamic behaviors of the proteins. In the form of complexes, the inhibitor chemicals emodin 8-o-b glucoside, hyperoside, penicillin, and resveratrol were studied using molecular dynamic simulations over a period of 100 ns. The ligand location was enhanced by a slight to and fro motion within the binding area (Figure 5). For the 100 ns simulations of the protein–ligand complex systems, the radius values of gyration were calculated to determine structural compactness. Comparable data were displayed throughout the simulation time frame, showing a real-time environment throughout the course of the experiment. During the simulation, systems remained compact, with no substantial conformational changes observed. The radius of gyration maps of the systems may be examined in Figure 7. This was followed by RMSF analysis, which showed the flexibility and stability of the residues. It was inferred that all the inhibitors displayed some structural conformation changes, but the ligands inside the cavity remained stable throughout the simulation time period. Thus, these computational approaches can enhance the ability of researchers to authenticate the therapeutic and prophylactic effects of these drugs during in vitro and in vivo studies.

## 4. Materials and Methods

### 4.1. Ligand Preparation and Selection

Ten different ligands extracted previously from *Reynoutria japonica* were selected based on molecular docking studies for investigation of their anti-quorum-sensing potential against MRSA, as shown in Table 1. These chemical compounds were retrieved from the PubChem database accessed on 5 October 2022 and were minimized using Chem Draw version 12.02 by applying an MMFF94 force field [38,39]. This was followed by the use of UCSF Chimera 1.14, via which compounds were again minimized using 500 steps of the steepest descent algorithm and 500 steps of the conjugate gradient to remove the rigidness and addition of hydrogens [40,41]. 

### 4.2. Bioactivity Analysis of Ligands and Toxicity Measurement

The potential success of a compound depends on its ADMET properties and the Lipinski rule of five. Swiss ADME accessed on 8 October 2022was applied to filtered out the best molecules according to the properties of drug-likeness and lead-likeness [42]. The drug-likeness rules used included the Lipinski rule of five (MW ≤ 500, HBA ≤ 10, MLogP ≤ 4.15, HBD ≤ 5, TPSA 40–130 Å^2^), Veber filter (rotatable bonds ≤ 10, TPSA ≤ 140), Ghose filter (LogP ≥ 0.4–≤ 5.6, MW ≥ 160–≤ 480, atoms ≥ 20–≤ 70, MR ≥ 40–≤ 130), Egan rule (WLogP ≤ 5.88, TPSA ≤ 131.6), and Muegge rule (TPSA ≤ 150, number of rings ≤ 7, number of carbons > 4, number of heteratoms > 1, HBA ≤ 10, MW ≥ 200–≤ 600, number of rotatable bonds ≤ 15, HBD ≤ 5, XLogP ≥ −2–≤ 5). Drug-like compounds were scrutinized further based on filtering of lead-likeness (250 ≤ MW ≤ 350, XLOGP ≤ 3.5, and rotatable bonds ≤ 7) [43]. The resultant set of inhibitors were then minimized using an MMFF94 force field in Chem Draw version 12.02 [22]. 

### 4.3. Target Protein Selection and Primary Sequence Retrieval

The functional roles of the proteins, catalytic needs for enzymatic activity, and active isoforms’ dependence on cofactors, subunit structure, and related post-translational changes were all retrieved from UNIPROTKB accessed on 10 October 2022 and the Protein Database (PDB) accessed on 10 October 2022 [44,45]. Additionally, the selected potential targets’ structural features were investigated to determine the accessibility of their experimental structures. The target proteins, which were selected on basis of their virulence and pathogenicity factors, were accessory gene regulator proteins A, B, and C and TRAP [46]. The primary sequences of the target proteins AgrA, AgrB, AgrC, and TRAP were retrieved in FASTA format from the protein sequence database UniProt (http://www.uniprot.org/ accessed on 10 October 2022), along with information about accession number and residue lengths [45]. Prior to docking, protein structures were prepared using the dock prep method in UCSF Chimera 1.14 [41]. The target proteins were then put through energy reduction to enhance their quality. Chimera, a potent visualization tool from UCSF, was employed to analyze the structures and reduce energy consumption [41]. Under the ff03.rl force field, 1500 rounds of a minimization run (750 steepest descent followed by 750 conjugate gradient) with a step size of 0.02 were used to assign Gasteiger charges to proteins and eliminate structural restrictions [47]. A validation technique was used to assess the protein minimization before the structures were applied in the docking investigations [48].

### 4.4. Physiochemical Properties and 3D Structures of Proteins

ProtParam (https://web.expasy.org/protparam/ accessed on 10 October 2022) was used to predict the properties of AgrA, AgrB, AgrC, and TRAP. The number of positively charged residues (Arg+Lys) and negative charged residues (Asp+Glu), theoretical pI, molecular weight, ext coefficient (Cys included), ext coefficient (Cys not included), instability index, aliphatic index, and grand average of hydrophobicity were computed using ProtParam [49]. The 3D structures were retrieved from PDB (https://www.rcsb.org/ accessed on 10 October 2022). I-TASSER (https://zhanglab.ccmb.med.umich.edu/I-TASSER/ accessed on 10 October 2022) was used as an alternative for some structures that were not available on PDB. Alphafold (https://alphafold.com/ accessed on 10 October 2022), an authentic protein structure database, was also used for the prediction of proteins’ 3D structures [50].

### 4.5. Structure Analysis and Functional Domain Identification 

PyMOL (https://pymol.org/ accessed on 20 October 2022) is a cross-platform molecular graphics tool that has been used worldwide for the three-dimensional analysis and visualization of many proteins and small molecules. After downloading the protein structures, the extra constituents attached to the proteins were removed using the open-source PyMOL system [19]. Interpro (http://www.interpro.com/ accessed on 25 October 2022), an online database, was used to identify the functional domains of the target proteins AgrA, AgrB, AgrC, and TRAP [51]. 

### 4.6. Active Site Identification

A ligand shows the maximum or highest interaction with the active site of its target protein. Amino acids are highly involved in the formation of ligand–protein complexes. Protein binding pockets were identified using CASTp software (http://sts.bioe.uic.edu/castp/ accessed on 5 November 2022) [52].

### 4.7. Molecular Docking of Targeted Proteins

The docking process was carried out using the minimized proteins along with the minimized ligand molecules. AutoDock Vina 4.2 [53] was used for the evaluation of docking and binding affinities. The best ligands were characterized on the basis of their binding affinities. To visualize the docked protein complexes and to understand in detail the interactions that contributed to the binding of ligands, Visual Molecular Dynamics (VMD) v1.93 [54], LIGPLOT [24], UCSF Chimera 1.16 [41], and Discovery Studio (DS) Visualizer 3.5 were used.

AutoDock Vina (accessed on 11 November 2022) is a docking program which docks a partial flexible ligand to a partial flexible protein. It implements a genetic algorithm and shows approximately 71% success in identifying docked ligand binding modes, the same as experimental identification [55]. The current study used an AutoDock Vina function which describes van der Waals interactions and hydrogen bonding in terms of energies.

### 4.8. Lead Compound Identification

After a detailed analysis of the physiochemical and pharmacokinetic properties of the proteins and ligands and the docking score comparison, the most active inhibitor was identified. The selected compound was the lead compound.

### 4.9. Reference Antibacterial Drug Identification and Selection

This step was performed for the identification of drugs that are used for the treatment of bacterial diseases. The Drug Bank (https://go.drugbank.com/ accessed on 25 December 2022) database was used for drug identification because it allowed us to analyze the drugs in detail along with their pathways [56].

### 4.10. Prediction of Different Parameters of Selected Drugs

The selected medications were filtered to identify the most efficient drug. This was done through a detailed study of the identified drugs and the most effective drug was identified using set parameters, i.e., physiochemical properties, effective ADMET properties, and mechanism of action and minimal side effects, which were determined using the PubChem, Drug Bank, and pkCSM databases, respectively. The identified drug was then docked with the target proteins to identify its inhibition efficiency. 

### 4.11. Reference Drug and Lead Compound Comparison

The comparison between the reference antibacterial drug and the proposed lead compound was done by comparing docking scores and physiochemical and ADMET properties.

### 4.12. Molecular Dynamic Simulations 

For structural analysis, MD simulations of the docked complexes were run for certain time periods using the AMBER program [57]. The Antechamber tleap interface was applied during system preparation and the preprocessing phase. The general AMBER force field (GAFF) [58] was applied for ligands, while the ff14SB force field was applied for the enzymes [59]. Using the LEaP module [60], the topologies of the enzymes and their inhibitors were recorded. To make each system electrostatically neutral, eight or nine Na + ions were added to the complexes as appropriate. Each system was inserted into the water molecule TIP3P box. This cubic box was employed in the simulations due to its geometric simplicity. The steepest descent approach was used for 1500 steps, followed by application of the conjugate gradient method for 1000 steps to minimize energy use. Each system was heated for 10 ps at a steady 300 K temperature and constant volume (canonical ensemble). Each system was then brought into equilibrium for 100 ps using Langevin’s thermostat and periodic boundary conditions with constant pressure. Explicit solvent models were used to execute the production runs for 10 ns for all systems in an isothermal–isobaric ensemble (T = 300 K; *p* = 1 atm). Periodic boundary conditions (PBCs) and the particle-mesh Ewald (PME) technique [61] were utilized to describe the long-range electrostatic effects, and a weak coupling algorithm was employed to relate the temperature to an external bath. Using the SHAKE method [34], the bond lengths, including of hydrogen bonds, were restricted. The Langevin coupling integration approach was used to maintain a consistent temperature. Newton’s equations were solved with a time step of 2 fs, and for the subsequent investigation, trajectory data were gathered every 1 ps. All MD trajectory studies were conducted using the Ptraj module of AmberTools 20, and visual inspection was done using VMD software v1.93.

## 5. Conclusions

The study confirmed that resveratrol had the best quenching abilities against the studied target proteins of *Staphylococcus aureus*. Continued observations and fundamental research on bacterial pathogenicity and intercellular signaling will aid in the creation of novel and effective treatments [49]. In pathogenic interactions, there will almost certainly be an ongoing conflict between microorganisms and their hosts. Because the anti-quorum-sensing tactics developed thus far have not yet been tested in large-scale clinical trials, it is difficult to evaluate their maximum potential and limitations at this point. However, we need to broaden our antimicrobial targets and approaches, and interference with intercellular signaling appears to be a viable and promising option for drug development. Medical studies are obviously required before more quorum-quenching-related products can become commercially available.

## Figures and Tables

**Figure 1 molecules-28-02635-f001:**
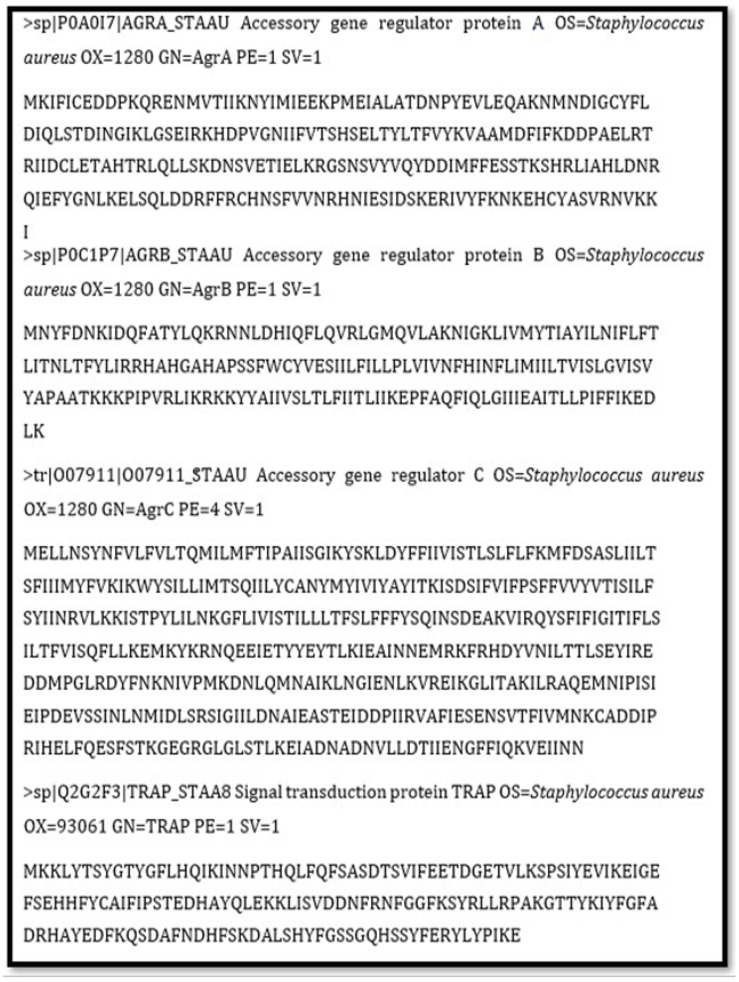
FASTA sequences of accessory gene regulator proteins A, B, and C and signal transduction protein TRAP.

**Figure 2 molecules-28-02635-f002:**
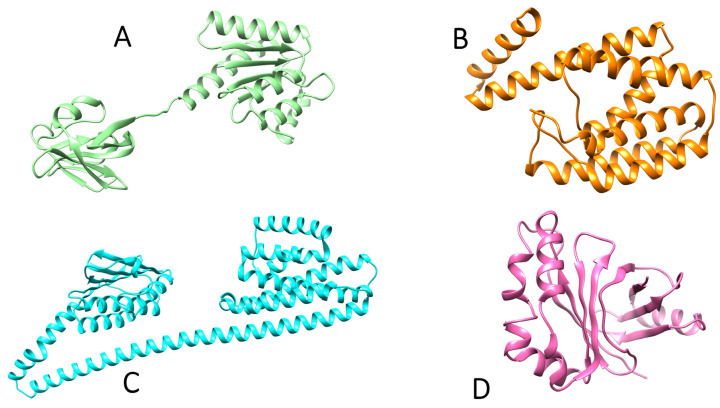
Structures of target proteins of *Staphylococcus aureus*: (**A**) AgrA, (**B**) AgrB, (**C**) AgrC, (**D**) TRAP. These structures were taken from Alphafold.

**Figure 3 molecules-28-02635-f003:**
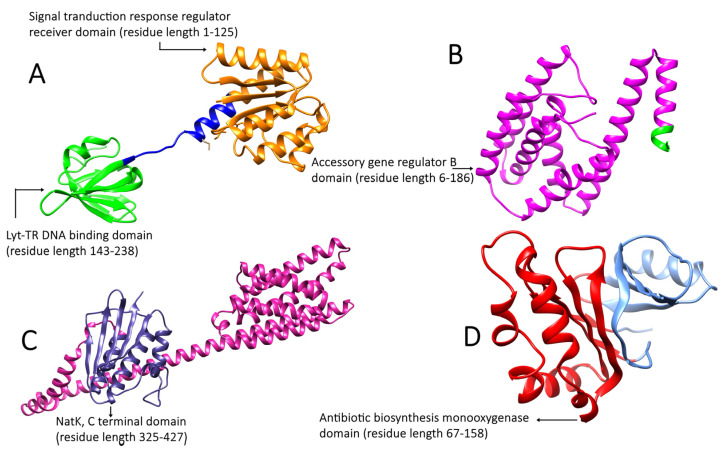
Functional domains of target proteins of *Staphylococcus aureus*. (**A**) AgrA: orange color showing the signal transduction response regulator receiver domain and green color showing the Lyt-TR DNA-binding domain; (**B**) AgrB: purple color showing the accessory gene regulator B domain; (**C**) AgrC: blue color showing the NatK C-terminal domain; (**D**) TRAP: red color showing the antibiotic biosynthesis monooxygenase domain.

**Figure 4 molecules-28-02635-f004:**
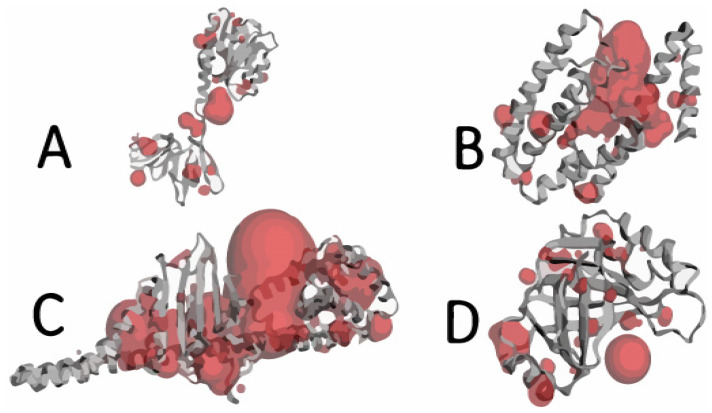
Binding pockets of target proteins of *Staphylococcus aureus*: (**A**) AgrA, (**B**) AgrB, (**C**) AgrC, (**D**) TRAP.

**Figure 5 molecules-28-02635-f005:**
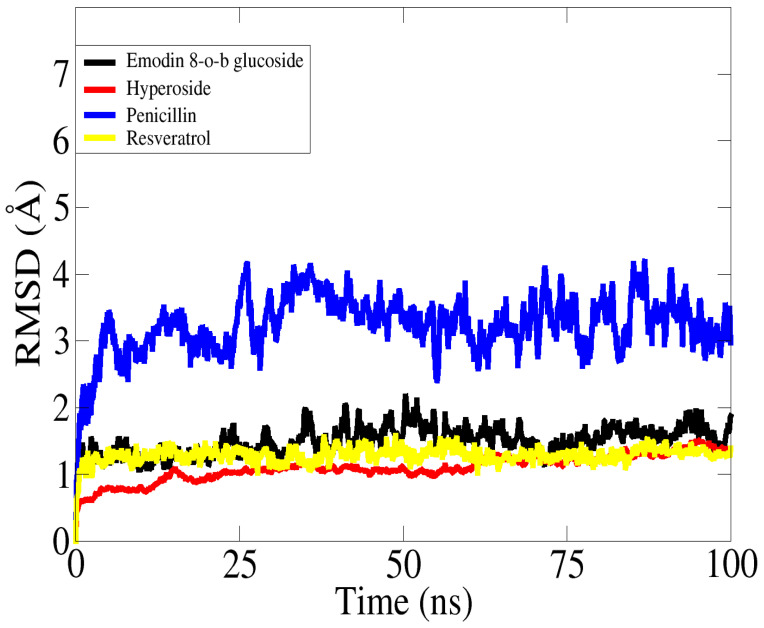
RMSD based on carbon alpha atoms of complexes.

**Figure 6 molecules-28-02635-f006:**
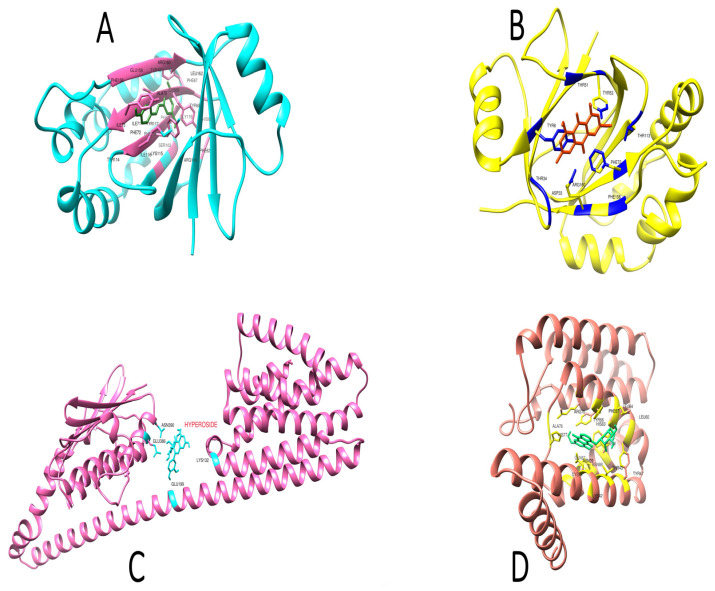
Simulated poses of the inhibitors against the target proteins. (**A**) Resveratrol, (**B**) emodin 8-o-b glucoside, (**C**) hyperoside complex, (**D**) penicillin.

**Figure 7 molecules-28-02635-f007:**
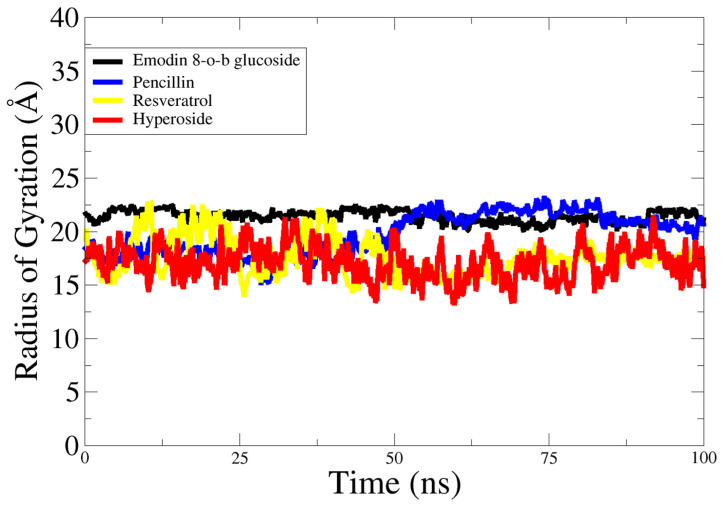
Radius of gyration analysis based on carbon alpha atoms of complexes.

**Figure 8 molecules-28-02635-f008:**
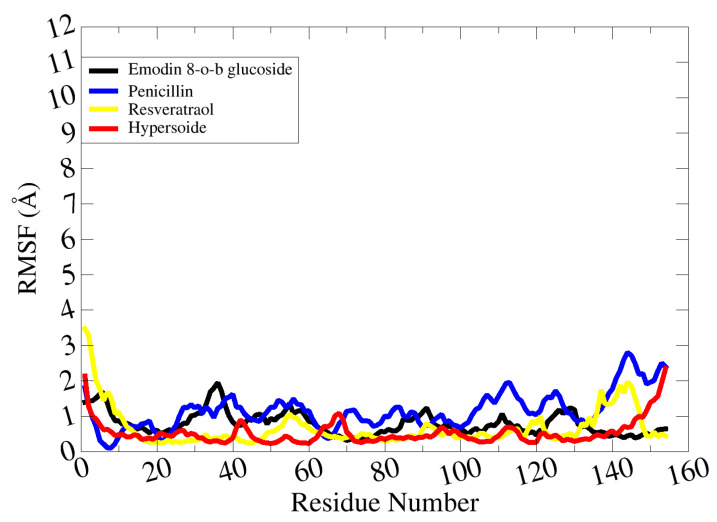
RMSF analysis of complexes.

**Table 1 molecules-28-02635-t001:** Physiochemical properties of target proteins.

Target Proteins	MW	PI	NR	PR	Ext.Co1	Ext.Co2	Instability Index	Aliphatic Index	GRAVY
AgrA	27,905.90	5.78	37	31	15,150	14,900	36.25	91.30	−0.379
AgrB	21,929.69	9.85	8	19	18,910	18,910	45.16	147.04	0.828
AgrC	49,896.91	5.19	45	38	38,405	38,280	39.15	127.16	0.494
TRAP	19,547.47	6.12	22	18	20,860	20,860	20.68	60.78	−0.580

**Table 2 molecules-28-02635-t002:** Structures of ligands with molecular formulas and molecular weights.

S. No	Ligand Name	Molecular Formula	Molecular Weight	Structure
1	2-methoxy-6-acetyl-7-methyljuglone	C_14_H_12_O_5_	260.24 g/mol	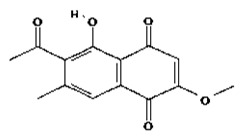
2	Emodin	C_15_H_10_O_5_	270.24 g/mol	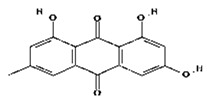
3	Emodin 8-o-b glucoside	C_21_H_20_O_10_	432.4 g/mol	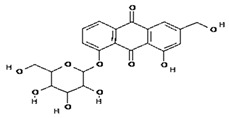
4	Polydatin	C_20_H_22_O_8_	390.4 g/mol	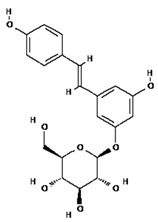
5	Resveratrol	C_14_H_12_O_3_	228.24 g/mol	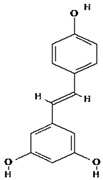
6	Physcion	C_16_H_12_O_5_	284.26 g/mol	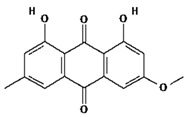
7	Citreorosein	C_15_H_10_O_6_	286.24 g/mol	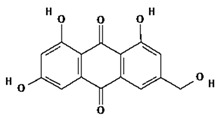
8	Quercetin	C_15_H_10_O_7_	302.23 g/mol	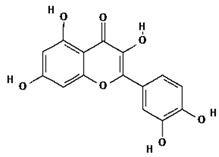
9	Hyperoside	C_21_H_20_O_12_	464.4 g/mol	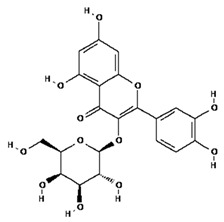
10	Coumarin	C_9_H_6_O_2_	146.14 g/mol	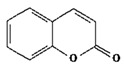

**Table 3 molecules-28-02635-t003:** The ligand molecules with binding score.

S. No	Ligand Name	Binding Score kcal/mol
1	2-methoxy-6-acetyl-7-methyljuglone	−7.1
2	Emodin	−8.4
3	Emodin 8-o-b glucoside	−9.9
4	Polydatin	−8.8
5	Resveratrol	−8.9
6	Physcion	−8.6
7	Citreorosein	−8.4
8	Quercetin	−8.8
9	Hyperoside	−9.1
10	Coumarin	−6.6

**Table 4 molecules-28-02635-t004:** Interactions of ligands with target proteins.

Ligands	Target Proteins with Interactive Residues
2-methoxy-6-acetyl-7-methyljuglone	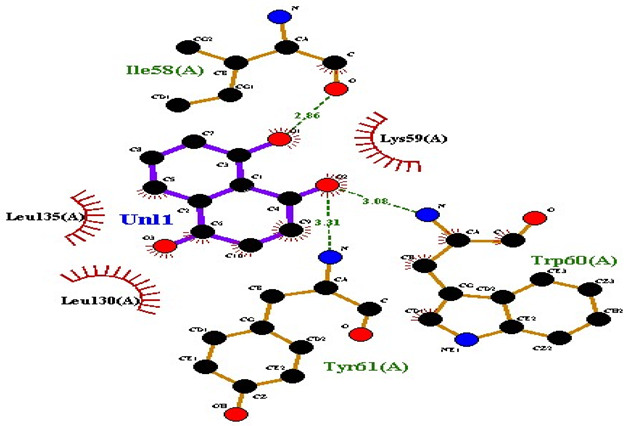 **AgrA**
Emodin	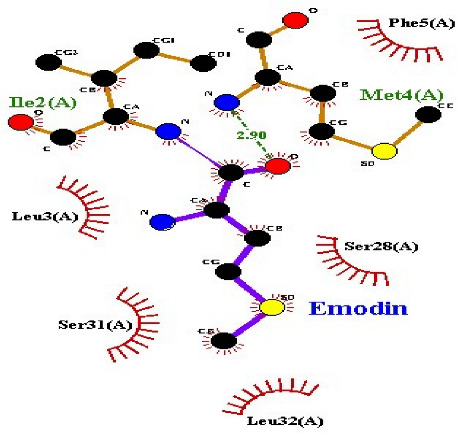 **TRAP**
Emodin 8-o-b glucoside	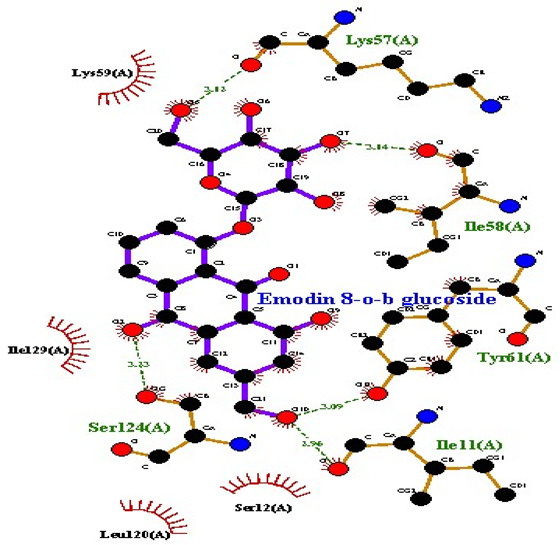 **AgrB**
Polydatin	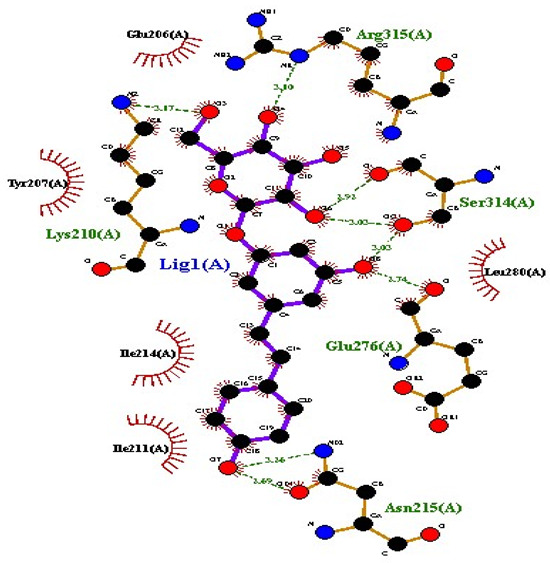 **TRAP**
Resveratrol	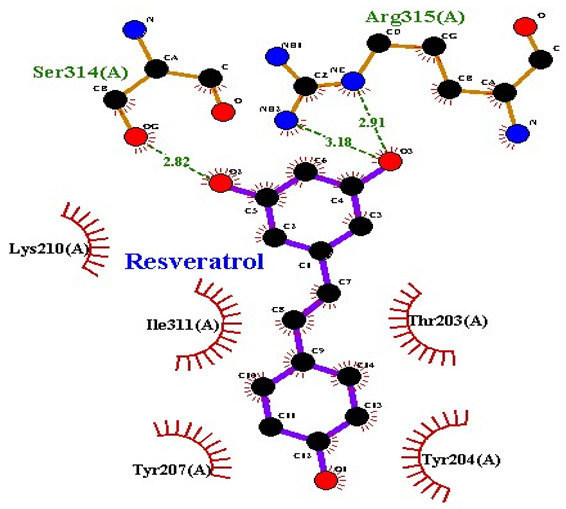 **TRAP**
Physcion	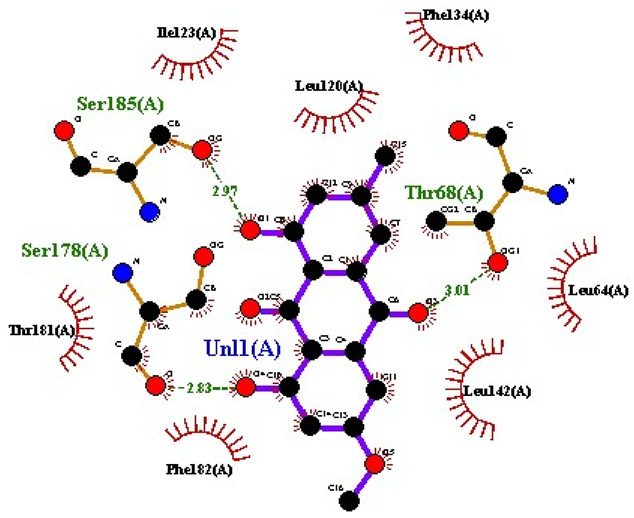 **AgrC**
Citreorosein	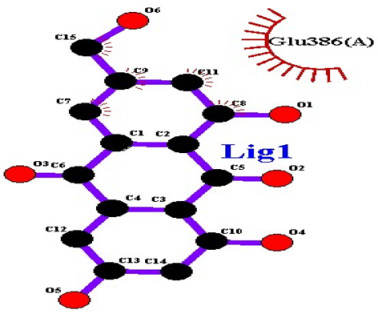 **AgrC**
Quercetin	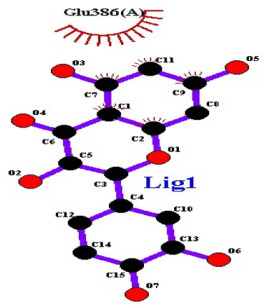 **AgrC**
Hyperoside	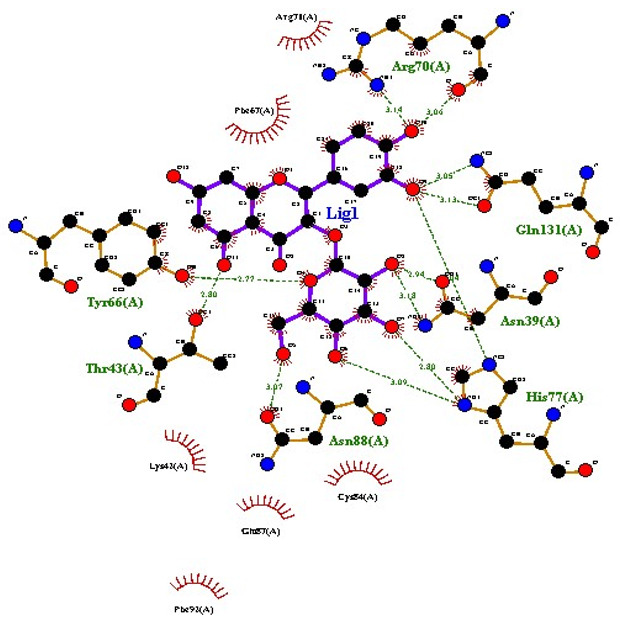 **AgrC**
Coumarin	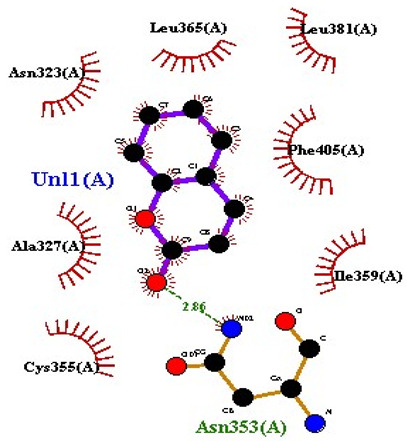 **AgrC**

**Table 5 molecules-28-02635-t005:** Amino acids, hydrogen bonding distances, and hydrophobic interactions.

S. No	Ligand Name	Binding Energy	No of HBs	Amino Acids	Hydrogen Bonding Distance	Hydrophobic Interactions
1	2-methoxy-6-acetyl-7-methyljuglone	−7.1	3	Ile58 Trp60 Tyr61	2.86 3.08 3.31	Lys59 Leu135 Leu130
2	Emodin	−8.4	2	Met4 Ile2	2.90 2.91	Phe5 Leu3 Ser28 Ser31 Leu32
3	Emodin 8-o-b glucoside	−9.9	5	Lys57 Ile58 Tyr61 Ile11 Ser124	3.12 3.14 3.09 2.96 3.23	Lys59 Ile129 Leu120 Ser12
4	Polydatin	−8.8	5	Arg315 Lys210 Ser314 Glu276 Asn215	3.10 3.17 3.03 2.76 3.26	Glu206 Tyr207 Leu280 Ile214 Ile211
5	Resveratrol	−8.9	2	Arg315 Ser314	2.91 2.82	Lys210 Ile311 Thr203 Tyr207 Tyr204
6	Physcion	−8.6	3	Ser185 Thr68 Ser178	2.97 3.01 2.83	Phe134 Leu120 Ile123 Leu64 Leu142 Thr181 Phe182
7	Citreorosein	−8.4	0	-	-	Glu386
8	Quercetin	−8.8	0	-	-	Glu386
9	Hyperoside	−9.1	7	Arg70 Gln131 Asn39 His77 Asn88 Thr43 Tyr66	3.06 3.13 2.94 2.80 3.02 2.80 2.22	Arg78 Phe67 Cys54 Lys43 Glu37 Phe92
10	Coumarin	−6.6	1	Asn353	2.86	Leu381 Leu365 Asn323 Phe405 Ile359 Ala327 Cys355

**Table 6 molecules-28-02635-t006:** Absorption properties of ligands.

S. No	Ligand Name	Water Solubility (mol/L)	CaCO_2_ Permeability (cm/S)	Intestinal Absorption (Human) %	Skin Permeability Log/Kp	P-Glycoprotein Substrate	P-Glycoprotein I Inhibtor	P-Glycoprotein II Inhibitor
1	2methoxy- 6-acetyl- 7-methyljugl one	−0.835	1.232	94.085	−2.77	No	No	No
2	Emodin	−3.271	0.259	71.316	−2.741	Yes	No	No
3	Emodin 8-o-b glucoside	−2.972	0.367	43.072	−2.735	Yes	No	No
4	Polydatin	−3.113	0.167	42.758	−2.735	Yes	No	No
5	Resveratrol	−3.235	1.196	87.933	−2.748	Yes	No	No
6	Physcion	−3.156	1.26	95.924	−2.8	Yes	No	No
7	Citreorosein	−3.186	−0.368	62.631	−2.74	Yes	No	No
8	Quercetin	−3.097	−0.277	76.081	−2.735	Yes	No	No
9	Hyperoside	−2.894	0.173	44.847	−2.735	Yes	No	No
10	Coumarin	−1.486	1.642	97.171	−1.911	Yes	No	No

**Table 7 molecules-28-02635-t007:** Applicability of Lipinski rule to ligands.

Ligands	Log*p* Value	Molecular Weight	H-Bond Acceptor	H-Bond Donor
Juglone	1.3274	174.155	3	1
Emodin	1.88722	270.24	5	3
Emodin 8-o-b	−1.1614	432.381	10	6
Polydatin	0.4469	390.388	8	6
Resveratrol	2.9738	228.247	3	3
Physcion	2.19022	284.267	5	2
Citreorosein	1.0711	286.239	6	4
Quercetin	1.988	302.238	7	5
Hyperoside	−0.5389	464.379	12	8
Coumarin	1.793	146.145	2	0

**Table 8 molecules-28-02635-t008:** Comparison of absorption properties.

S. No	Compound Name	Water Solubility (mol/L)	CaCO_2_ Permeability (cm/S)	Intestinal Absorption (Human) %	Skin Permeability Log/Kp	P-Glycoprotein Substrate	P-Glycoprotein Inhibitor	P-Glycoprotein II Inhibitor
1	Penicillin	−2.199	0.293	58.344	−2.735	Yes	No	No
2	Resveratrol	−3.233	1.196	87.933	−2.748	No	No	No

**Table 9 molecules-28-02635-t009:** Comparison of distribution properties.

S. No	Compound Name	VDss (Human) (L/kg)	Fraction Unbound (Human) (Fu)	BBB Permeability (Human) (Log BB)	CNS Permeability (Log PS)
1	Penicillin	−1.681	0.32	−0.741	−2.936
2	Resveratrol	0.022	0.089	−0.152	−2.113

**Table 10 molecules-28-02635-t010:** Comparison of metabolic properties.

Compound Name	CYP-2D6 Substrate	CYP-3A4 Substrate	CYP-2D6 Inhibitor	CYP-2619 Inhibitor	CYP-269 Inhibitor
Penicillin	No	Yes	No	No	No
Resveratrol	No	Yes	Yes	No	No

**Table 11 molecules-28-02635-t011:** Comparison of excretion properties.

S. No	Compound Name	Total Clearance (mL/Kg)	Renal OCT2 Substrate
1	Penicillin	0.02	No
2	Resveratrol	0.094	No

**Table 12 molecules-28-02635-t012:** Toxicity comparison.

S. No	Toxicity Parameters	Penicillin	Resveratrol
1	Max tolerated dose (human) (mg/kg)	1.284	0.561
2	hERGI inhibitor	No	No
3	hERGII inhibitor	No	No
4	Oral rat acute toxicity (mol/kg)	2.04	2.216
5	Oral rat chronic toxicity (mg/kg)	2.63	1.761
6	Hepatoxicity (log μg/L)	Yes	No
7	Skin sensitization	No	No
8	*T. pyriformis* activity (log μg/L)	0.285	0.982
9	Minnow toxicity (log mM)	2.255	1.367

**Table 13 molecules-28-02635-t013:** Penicillin and resveratrol: Lipinski rule of five.

S. No	Compound Name	Log*p* Value	Molecular Weight	H-Bond Acceptor	H-Bond Donor
1	Penicillin	0.8608	334.397 g/mol	4	2
2	Resveratrol	2.9738	228.247 g/mol	3	3

**Table 14 molecules-28-02635-t014:** Comparison of docking results.

S. No	Compound Name	Binding Score	Cavity Size	Grid Map	Minimum Energy (Kcal/mol)	Maximum Energy (Kcal/mol)
1	Penicillin	−6.7	86	23	0.00	1.6 × 10^0^
2	Resveratrol	−8.9	1857	34	0.00	1.6 × 10^0^

## Data Availability

The text and its supporting information files contain all pertinent data.

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
