# Peer review of "Quorum Quenchers from Reynoutria japonica in the Battle against Methicillin-Resistant Staphylococcus aureus (MRSA)"

_molecules, 2023, doi:10.3390/molecules28062635_

Round 1

Reviewer 1 Report

MS entitled, ‘Quorum Quenchers from Reynoutria japonica in the Battle of Methicillin Resistant Staphylococcus aureus (MRSA)’ is a good piece of works by authors to the ‘Molecules’. This study focuses on an alternative approach by using juglone derivatives from Reynoutria japonica as quorum quenchers. Ten bioactive compounds from this plant i.e., 2-Methoxy-6acetyl-7-methyljuglone, emodin, emodin 8-o-b glucoside, polydatin, resveratrol, physcion, citreorosein, quercetin, hyperoside and coumarin were taken as ligands and docked with accessory gene regulator protein  A, B, C and signal transduction protein TRAP. I have following suggestions:

# Authors are suggested to provide a schematic explaining the mechanism behind Quorum Quenching.

#  I found result section is poorly elaborated. Please elaborate with full details on selected proteins? Resolutions? Chain?

# table 2, authors didn’t take much efforts to make numbers at subscripts for said molecular formulaes of compounds. Do revise please.

#  At many places abbreviations are missing.

# I suggest authors to carry out suggested in-vitro analysis suggesting the Quorum Quenching activity.

#  Improve Table 4 figures.

# At many places authors have forgotten to fully abbreviate the said short forms. Authors should disclose them whenever first used. At many places authors forgotten to give date of accessions of webservers they used. e.g., pkSCM.

# I suggests its major revision.

Author Response

Response to Reviewer Comments

We thanks the Referees for spending their time and interest in our work. We have checked all the comments and have made necessary changes accordingly.

-Reviewer 1

1) I found result section is poorly elaborated. Please elaborate with full details on selected proteins? Resolutions? Chain?

Author Response: Thankyou for the comment. We acknowledge this comment and the result section regarding protein targets have been elaborated in line 111-124, 153-160, 177-183 with full details along the resolutions and chains in the revised manuscript.

2) table 2, authors didn’t take much efforts to make numbers at subscripts for said molecular formulaes of compounds. Do revise please.

Author Response: Thank you for highlighting this. The above comment has been adjusted accordingly. The numbers at subscript for the said molecular formulas of compounds at table 2 has been formatted accordingly.

3) I suggest authors to carry out suggested in-vitro analysis suggesting the Quorum Quenching activity.

Author Response: Thank you for the comment. Yes, we have carried out in-vitro analysis suggesting the Quorum Quenching activity, but those experimental results are accompanying one of another article.

4) Improve Table 4 figures.

Author Response: This comment has been acknowledged and the table 4 has been improved in the revised manuscript.

5) At many places authors have forgotten to fully abbreviate the said short forms. Authors should disclose them whenever first used. At many places authors forgotten to give date of accessions of webservers they used. e.g., pkSCM.

Author Response: We appreciate the reviewer for a valuable comment. The said comment has been addressed with full abbreviation of the said short forms. And have been disclosed in the first used.

6) At many places abbreviations are missing.

Author Response: Thanks for highlighting this. The said comment has been addressed accordingly.

Reviewer 2 Report

The study of Fatiha et al. has a good execution in the molecular modeling against MRSA biofilm formation and the health problems it represents. The article is well-detailed regarding the methods applied.

Understandably, the special issue is about computational methods; however, it is recommended that authors cite more references to justify the metabolite screening in PubChem since in vitro tests are not present that probes that resveratrol of the plant extract decreases absorbance of biofilm such as crystal violet assay. Is any publication on resveratrol against MRSA that could be cited, or some about EPS production or metabolic activity with MTT?

Are authors sure that resveratrol is a major component in Reynoutria japonica? References of isolation with HPLC or GC-MS could reinforce the hypothesis. If resveratrol is not a major component, why was this plant selected instead of another probed to have it? Extraction with solvents has a major impact on this type of polyphenol. It is recommended that a reference in which an in vitro test with R. japonica was tested could be included; otherwise, the research could be conducted with another species previously found to have resveratrol in high concentrations. The title could be modified to highlight the necessity of analyzing metabolites present in this plant, and not only towards the resveratrol properties since it could be confusing.

Reference 11 in line 94 does not mention the activity mentioned by the authors; please recheck this reference.

Table 1 does not have a title, and it seems to be incomplete.

In figure 2, the bacteria name is not italicized (which occurs throughout the paper). In the figure, letters appear as capital letters, but in the figure description, they are lowercase. The same comments apply to figures 3 and 4. Also, check the final point in the figure names. Improve figure resolution.

Table 2, structures lack resolution, and functional groups cannot be distinguished. The title name does not have a final point. In the molecular formula, numbers are not in subscripts.

Table 3 is mentioned in section 2.5 but appears until section 2.6.

In table 3, ligands' interaction with target protein structures needs to be better visualized due to lousy resolution. Authors should include better-resolution 3D models.

Text in sections 2.14 and 2.15 are not within the specified margins.

Table 5 headlines need to be better edited; it is suggested to use Excel formats for all tables to present them with more uniformity and alignment so that each headline is best understood.

Why was only penicillin used for comparison? If only one negative control was used for docking, adding a positive control from the stilbene group or even gallic acid is recommended as a reference.

In figure 5 for RMSD, it is recommended to include the RMSF analysis.

Line 390, "This plant was preferred because it showed miraculous anti-pathogenic activities against several diseases," is best to change the word "miraculous."

The discussion could have been easier to understand, and much text needed to be more relevant since it was introduction-like. The authors should discuss more findings.

References need to be updated (none from 5 years ago, at least).

This paper shows laborious work, but since the relevance or contribution to intrahospital and food safety problems it looks for, authors should be more precise about these topics or even include in vitro tests. At the same time, some methodologies could be omitted and give space to laboratory assays, or at least compared with other phenolic compounds from the stilbene family with MRSA biofilm formation inhibition activity. Try to improve writing and justification.

Finally, through all text, several details with periods, scientific names that are not in italic format, table titles, and subtitles without final points, and figure 7 not being referenced, among others, were found, for which it is that we ask to check the entire document and improve its quality cautiously.

Author Response

Response to Reviewer Comments

We thanks the Referees for spending their time and interest in our work. We have checked all the comments and have made necessary changes accordingly.

-Reviewer 2

1) Understandably, the special issue is about computational methods; however, it is recommended that authors cite more references to justify the metabolite screening in PubChem since in vitro tests are not present that probes that resveratrol of the plant extract decreases absorbance of biofilm such as crystal violet assay. Is any publication on resveratrol against MRSA that could be cited, or some about EPS production or metabolic activity with MTT?

Author Response:  Resveratrol has a good potential as antioxidant and antibacterial and reported in several studies.

t was proposed the antibiofilm activity of resveratrol might be due to its ability to cause interference in quorum sensing thus affecting synthesis of surface proteins and capsular polysaccharides—downregulation of genes (cap5ABCFG) responsible for capsular polysaccharide synthesis was reported with resveratrol treatment at 100 μg/mL. 

Qin, N., Tan, X., Jiao, Y., Liu, L., Zhao, W., Yang, S., et al. (2014). RNA-Seq-based transcriptome analysis of methicillin-resistant Staphylococcus aureus biofilm inhibition by ursolic acid and resveratrol. Sci. Rep. 4:5467. doi: 10.1038/srep05467

2) Are authors sure that resveratrol is a major component in Reynoutria japonica? References of isolation with HPLC or GCMS could reinforce the hypothesis. If resveratrol is not a major component, why was this plant selected instead of another probed to have it? Extraction with solvents has a major impact on this type of polyphenol. It is recommended that a reference in which an in vitro test with R. japonica was tested could be included; otherwise, the research could be conducted with another species previously found to have resveratrol in high concentrations. The title could be modified to highlight the necessity of analyzing metabolites present in this plant, and not only towards the resveratrol properties since it could be confusing.

Author Response: Rjaponica rhizome is a major source of resveratrol for the dietary supplement industry.

Alperth, F., Melinz, L., Fladerer, J. P., & Bucar, F. (2021). UHPLC Analysis of Reynoutria japonica Houtt. Rhizome Preparations Regarding Stilbene and Anthranoid Composition and Their Antimycobacterial Activity Evaluation. Plants10(9), 1809.

The compounds are already extracted, purified and validated for anticancerous activities and antibacterial against H. pylori however their potential as antibiofilm assay is the current focus as this approach will stop the signaling without putting any selective pressure on the bacterial cells. 

Kim, J. H., Khalil, A. A. K., Kim, H. J., Kim, S. E., & Ahn, M. J. (2019). 2-Methoxy-7-Acetonyljuglone isolated from reynoutria japonica increases the activity of nuclear factor erythroid 2-related factor-2 through inhibition of ubiquitin degradation in HeLa cells. Antioxidants8(9), 398.

Khalil, A. A. K., Park, W. S., Lee, J., Kim, H. J., Akter, K. M., Goo, Y. M., ... & Ahn, M. J. (2019). A new anti-Helicobacter pylori juglone from Reynoutria japonica. Archives of pharmacal research42, 505-511.

3) Reference 11 in line 94 does not mention the activity mentioned by the authors; please recheck this reference.

Author Response: Thank you for highlighting this mistake.  Reference 11 in line 94 has been revisited and addressed properly.

4) Table 1 does not have a title, and it seems to be incomplete"

Author Response: Thank you for highlighting this mistake. The title have been added and has been corrected.

5) In figure 2, the bacteria name is not italicized (which occurs throughout the paper). In the figure, letters appear as capital letters, but in the figure description, they are lowercase. The same comments apply to figures 3 and 4. Also, check the final point in the figure names. Improve figure resolution.;

Author Response: Thank you for your valuable comment. The figure 2 has been revised completely, bacteria name is italicized. The figure letters description and the resolution of figure 2, 3 and 4 has been revisited and made it with high resolution in the revised manuscript.

6) Table 2, structures lack resolution, and functional groups cannot be distinguished. The title name does not have a final point. In the molecular formula, numbers are not in subscripts.;

Author Response: Thanks for the comment. Yes the mentioned Table 2 and all the other structures resolution has been revisited and adjusted properly in the revised manuscript.

7) Table 3 is mentioned in section 2.5 but appears until section 2.6";

Author Response: Thank you for highlighting this mistake. Proper arrangement of table 2.6 has been adjusted in the revised manuscript.

8) In table 3, ligands' interaction with target protein structures needs to be better visualized due to lousy resolution. Authors should include better-resolution 3D models.

Author Response: We acknowledge your comment. The table 3 targeted protein structure has been readjusted and proper target names with better resolution has been revised in the manuscript.

9) Text in sections 2.14 and 2.15 are not within the specified margins.

Author Response: Thank you for highlighting this mistake. Text in section 2.14 and 2.15 have been formatted accordingly in the revised manuscript.

10) Table 5 headlines need to be better edited; it is suggested to use Excel formats for all tables to present them with more uniformity and alignment so that each headline is best understood.

Author Response: The authors acknowledge the comment, and the table 5 headlines has been addressed properly in the revised manuscript.

11) Why was only penicillin used for comparison? If only one negative control was used for docking, adding a positive control from the stilbene group or even gallic acid is recommended as a reference.

Author Response: Penicillin is a narrow spectrum antibiotic used against gram positive bacteria, penicillin was used as reference dug to check any difference in activity as MRSA produces enzymes penicillinases to inactivate the basic or core ring of antibiotic.

12) In figure 5 for RMSD, it is recommended to include the RMSF analysis.

Author Response: Yes, the above comment has been acknowledged and the RMSD has been added to page 26 at the revised manuscript.

13) Line 390, "This plant was preferred because it showed miraculous anti-pathogenic activities against several diseases," is best to change the word "miraculous."

Author Response: Thanks for highlighting the mistake the word "miraculous" has been change to “exceptional”

14) The discussion could have been easier to understand, and much text needed to be more relevant since it was introduction-like. The authors should discuss more findings.

Author Response: Thank you for the comment. The comment has been addressed properly with more findings.

15) References need to be updated (none from 5 years ago, at least

Author Response: Yes, we have updated the reference in the revised manuscript.

20) This paper shows laborious work, but since the relevance or contribution to intrahospital and food safety problems it looks for, authors should be more precise about these topics or even include in vitro tests. At the same time, some methodologies could be omitted and give space to laboratory assays, or at least compared with other phenolic compounds from the stilbene family with MRSA biofilm formation inhibition activity. Try to improve writing and justification.

Author Response: Thank you for the useful suggestions. As the study is pure computational and the research team domain is not experimental, the main objective of the study was to present in silico findings to the research community. However, we are collaborating with experimentalists to do the biological activities of the screened compounds in the near future.  

21) Finally, through all text, several details with periods, scientific names that are not in italic format, table titles, and subtitles without final points, and figure 7 not being referenced, among others, were found, for which it is that we ask to check the entire document and improve its quality cautiously.

Author Response: Yes, we acknowledge the suggestion. The said comment has been revisited and addressed properly. Typos mistakes, format of the scientific notation, table titles, subtitles and figure 7 has cited properly.   

Round 2

Reviewer 1 Report

Authors have corrected successfully

Reviewer 2 Report

Thanks for attending all the observations and extensive editing.